# Modern and Non-Invasive Methods of Fat Removal

**DOI:** 10.3390/medicina59081378

**Published:** 2023-07-28

**Authors:** Patrycja Piłat, Gabriela Szpila, Michał Stojko, Jakub Nocoń, Joanna Smolarczyk, Karol Żmudka, Martyna Moll, Michał Hawranek

**Affiliations:** 1Student’s Scientific Society, Department of Psychiatry, Faculty of Medical Sciences in Zabrze, Medical University of Silesia, 40-055 Katowice, Poland; s78778@365.sum.edu.pl (G.S.); s83148@365.sum.edu.pl (M.S.); s83022@365.sum.edu.pl (J.N.); s78586@365.sum.edu.pl (K.Ż.); martynamoll@gmail.com (M.M.); 2Student’s Scientific Society, III Department of Cardiology, Faculty of Medical Sciences in Zabrze, Medical University of Silesia, 40-055 Katowice, Poland; 3Department of Psychiatry, Faculty of Medical Sciences in Zabrze, Medical University of Silesia, 50-055 Katowice, Poland; joanna.smolarczyk@sum.edu.pl; 4III Department of Cardiology, Faculty of Medical Sciences in Zabrze, Medical University of Silesia, 40-055 Katowice, Poland; mhawranek@poczta.fm

**Keywords:** lipolysis, modern methods, obesity, innovation, fat tissue

## Abstract

Adipocytes accumulate triacylglycerols as an energy store, thereby causing an increase in the adipose tissue volume. Weight gain can be prevented through damage to the adipocyte structure or an increase in the body’s metabolic rate. Commonly used methods to disintegrate the cell membrane of adipocytes include injection lipolysis, cryolipolysis, ultrasonic lipolysis, radiofrequency lipolysis, laser lipolysis, carboxytherapy, and lipolysis using an electromagnetic field. The names of these methods suggest which substances are being used, and their main advantages are a very low invasiveness, as well as effectiveness. However, new discoveries in medicine, along with individuals’ desire to improve their appearance, have resulted in numerous studies on more ways of reducing body fat. Great potential is seen in beige adipocytes, which can be transformed, i.e., “recruited” from white adipocytes, or synthesized de novo; they also show thermogenic properties. One of the stimuli inducing the formation of beige adipocytes is cold and B3-adrenergic stimulation. Based on these findings, the researchers created, for example, cooling clothing. Additionally, curcumin and natural anthocyanins have proven to be helpful in the treatment of obesity and diabetes, by stimulating the secretion of glucagon-like peptide-1, and inducing the formation of beige adipocytes. Another study showed that the conversion of white adipose tissue is indirectly influenced by interleukin-6 secreted by the muscles, the expression of which is increased in people actively exercising. Moreover, there is potential in adenosine analogs, fenoldopam, rhubarb, the herbal extract *Ephedra sinica* Stapf, electroacupuncture simulation, and the drug CBL-514. Despite knowledge and experience, the ideal method for a quick and noticeable, but safe and non-invasive reduction of body fat has not been found yet. The research conducted nowadays may bring us closer to the development of a universal method, and turn out to be a breakthrough in the fight against overweight and obesity.

## 1. Introduction

Obesity (body mass index, BMI ≥ 30) and overweight (BMI > 25) have always been a serious problem in society, and a challenge for healthcare professionals, mainly due to the complications caused by an increased body weight. The World Health Organization (WHO) in 2022 released a report, taking into account the period of the COVID-19 pandemic, showing that most European countries are struggling with a rapidly growing rate of overweight, including among young people. According to recent publications, obesity and overweight rank fourth as risk factors for non-communicable diseases. Almost 60% of adults, and about 30% of children are overweight or obese. An alarming fact is that in Europe, as many as 7.9% of children under the age of five are obese or overweight, and Poland is ranked 15th in this ranking [1]. There are many reports that the COVID-19 pandemic has caused weight gain, mainly among young people and children [2,3].

Obesity is defined as an abnormal or excessive fat accumulation that may impair health. Adipose tissue, otherwise known as body fat, is a type of connective tissue, in which fat storage cells predominate. It consists mainly of cells called adipocytes, which are formed from fibroblast-like, progenitor, undifferentiated cells, known as preadipocytes [4]. Adipose tissue accounts for about 15–20% of body weight in men, and slightly more in women [5]. The fundamental role of the adipocyte is the synthesis and storage of energy in the form of triglycerides, i.e., simple fats [6].

Brown adipose tissue (BAT) plays a major role in energy homeostasis and thermogenesis among mammals. It prevents obesity caused by diet and hypothermia, through uncoupling protein 1 (UCP1). This mechanism is very important in young children, due to the undeveloped capacity of thermoregulation [4]. Furthermore, BAT occurs among adults in the cervical–supraclavicular areas, the nape, between the scapulae, along the spinal cord, close to the large vessels in the mediastinum, and around the cardiac apex. The BAT activity correlates positively with the increase in energy expenditure during exposure to cold, and negatively with age, body mass index, and fasting glucose [7,8]. The UCP-1 protein generates heat instead of ATP, as a result of mitochondrial catabolic activity. The process is induced by adrenergic stimulation, thyroxine, and staying at a low temperature [8]. Some adipocyte precursors and WAT adipocytes are capable of the de novo synthesis or direct conversion, as well as the transdifferentiation, of the so-called “browning”. The last refers to the process where, in response to stimuli, including a prolonged exposure to cold, the effects of β3-adrenergic receptor (ADRB3) agonists (here: catecholamines), peroxisome proliferator-activated receptor gamma (PPARγ, here: fibrates and thiazolidinediones), physical exercise, and cachexia, beige adipocytes appear among the WAT cells. These cells have features morphologically and biochemically similar to the brown ones, due to the increased number of active mitochondria, the reduced volume of lipid droplets, the increased oxidation of fatty acids, and the uncoupling of the respiratory chain [4,9,10]. 

Pink adipose tissue (PAT) is formed during pregnancy and lactation from the transformation of subcutaneous WAT into a specific female cell type, the so-called “pink adipocyte”. The pink adipocyte is a mammary gland alveolar epithelial cell, whose role is to produce and secrete milk, formed by lipid-rich elements. After the period of lactation, these cells are converted back into white adipocytes, and the energy needs of the woman return to the baseline level [11,12]. 

The number of fat cells during a person’s lifetime remains relatively constant, regardless of BMI or weight loss [13]. Therefore, the number of adipocytes is the main determinant of fat mass among adults, because the cells formed in childhood do not disappear over the years, but have the ability to adapt by increasing in volume as a result of lipid accumulation, or decreasing when there is a negative energy balance [14]. Nowadays, more than 40% of the calories in a typical diet of a modern human in a developed country are supplied by fat, and a similar amount by carbohydrates, which, when supplied in excess, are converted into triglycerides [4]. Under the influence of lipases, triglycerides are broken down into fatty acids and glycerol, and these are then transported with the blood to currently active tissues, where their oxidation takes place and, as a result, energy is generated in the form of ATP or heat [4,5]. So far, obesity prevention and treatment strategies, both at the individual and population level, have not been successful in their long-term effects [15]. New treatments focus on the elimination of adipocytes by destroying their structure and inducing apoptosis. 

The research combining the above methods is currently underway, and may lead to more comprehensive and synergistic effects than the use of monotherapy. An increased adipose tissue mass is the basic phenotypic feature of obesity.

## 2. Current Methods

Presently, there are five leading non-invasive or micro-invasive techniques that allow us to reduce fat cells; i.e., injection lipolysis, cryolipolysis, ultrasonic lipolysis, radiofrequency lipolysis, and laser lipolysis. 

### 2.1. Cryolipolysis

Cryolipolysis uses cold to freeze the fat cells sucked by the device in the targeted areas. It causes the apoptosis of adipocytes, which are then naturally absorbed and eliminated. During the procedure, patients may feel the skin pulling and pinching, but the effects are not immediately visible [16,17]. The principle of cryolipolysis is based on the assumption that adipocytes are more susceptible to cooling than other skin cells. Their death is initiated when they are cooled to −1 °C. To be more specific, an inflammatory response is induced, inflammatory cells flow in and, within 2 weeks, the adipocytes are surrounded by histiocytes, neutrophils, lymphocytes, and other mononuclear cells. Then, the macrophages surrounding the adipose tissue reduce the inflammatory focus, by metabolizing the lipid cells. About a month after the procedure, the inflammation and the volume of the adipocytes decrease [18,19]. Histological analysis 2–3 months after cryolipolysis shows a significant reduction in the distance between the adipose tissue septa, and no changes in the lipid profile. Just one session leads to a reduction in the thickness of body fat by up to 20%, and the effects are visible after 3–4 months. Cryolipolysis is known for its good efficacy and safety profile; the side effects reported so far are insignificant [20].

### 2.2. Laser Lipolysis

Laser lipolysis has been a commonly used method to reduce body fat since 2006. The generated laser beam, with possible wavelengths of 924, 968, and 980 nm for the diode laser, and 1064, 1319, 1320, and 1440 nm for the Nd:YAG laser (neodymium-YAG laser) penetrates the tissue, and is scattered, reflected, and absorbed [21]. The laser fat reduction depends on the laser wavelength and energy. The principle of this non-invasive method is based on raising the temperature of adipocytes to the threshold of cell death, i.e., 42~47 °C; their complete decomposition occurs at 50~65 °C. With the accumulation of heat, the adipocytes are irreversibly damaged by the formation of transient micropores in their membranes, followed by the release of intracellular lipids. As with cryolipolysis, the inflammation leads to the removal of dead cells [21,22]. Innovative devices for laser lipolysis are equipped with a cooling system, or supported by a cryogenic spray, to protect against the side effects of laser photothermolysis, and improve the general therapeutic effect [22,23]. Laser therapy has been proved to be effective in reducing the waist, hip, and thigh circumference, closing small vessels, removing skin imperfections (e.g., rosacea and erythema) and signs of skin aging such as a loss of firmness and tissue flabbiness (e.g., wrinkles and folds), but also in lowering cholesterol and leptin levels, while remaining a non-invasive and safe method [24].

### 2.3. Injection Lipolysis 

Injection lipolysis is often confused with mesotherapy; however, injection lipolysis is a much broader concept. Lipolysis describes the hydrolysis or degradation of lipids into their constituent fatty acids and glycerol esters. This process, regulated by lipases, takes place in the adipocytes and vascular space of the muscle or adipose tissue [25]. In treatment aimed at reducing body fat, mesotherapy is used, which constitutes multiple intradermal injections of small doses of the appropriate drug into selected parts of the body. However, injection lipolysis is most often associated with the technique of 1–2 punctures, such as intralipotherapy, repeated in cycles of 2–5 series at intervals of 4–6 weeks. The active substances applied for the subcutaneous fat are most commonly phosphatidylcholine (PPC) or sodium deoxycholate (DC) [26,27,28]. PPC is the basal phospholipid component of cell membranes, and a precursor to acetylcholine [29]. As a result of the subcutaneous injection of PPC into the adipose tissue, adipocytes burst, and the phosphatidylcholine increases the secretion of triglyceride-rich lipoproteins. DC is a component of human bile acid, which helps in the emulsification and digestion of fats in the intestines. However, when administered externally, it damages the cell membrane of adipocytes, causing their death [27]. The best-known combinations of these two drugs in injection lipolysis are Lipostabil PPC (50 mg/mL), along with DC, sodium hydroxide, sodium chloride, α-tocopherol, benzyl alcohol and ethanol, and formulations differing in the value of DC, i.e., Kybella, which contains DC (10 mg/mL), and GeoLysis, which contains DC (10 mg/mL). The cascade of inflammatory response triggered by the adipocyte destruction causes the migration and stimulation of fibroblasts and, eventually, the accumulation of more collagen [30]. 

### 2.4. Ultrasonic Lipolysis

Ultrasonic lipolysis and, more specifically, high-intensity focused ultrasound (HIFU) lipolysis, affects the adipose tissue by two mechanisms. It causes negative acoustic pressure that breaks cell membranes by inducing cavitation bubbles, releasing heat energy, and increasing the temperature, which causes protein denaturation and coagulation [31,32,33]. The cavitation phenomenon used here affects the adipose tissue by converting it into a liquid form, which is then naturally excreted in urine. A higher temperature is achieved through ultrasonic vibrations [34]. The effect is the destruction of adipocytes, with the release of lipids, which are transported to the hepatobiliary system through lymphatic drainage. The result of these mechanisms is a reduction in the local volume of adipose tissue. This method is completely non-invasive and painless [35].

### 2.5. Radiofrequency Lipolysis

Radiofrequency lipolysis uses radio waves that emit heat, which is selectively directed to a layer of collagen-rich tissue, causing the shrinkage and denaturation of collagen fibers, and the stimulation of fibroblasts. This method is mainly used to firm the skin and gently reduce body fat through the thermal apoptosis of adipocytes [16,17]. It works best as an additional therapy for other fat-removal procedures [36]. 

### 2.6. Low-Level Laser Therapy

Low- level laser therapy (LLLT), emitting waves between 630–640 nm, can be used as a supplement to lipoplasty. The EML laser, which emits 14 mW of light with a wavelength of 635 nm, is applied to the surface of the skin before the liposuction procedure. This allows the fat to emulsify, and softens the body area prior to suction. This therapy shortens the procedure time, increases the volume of fat removed, and accelerates the patient’s recovery [37]. In addition, this therapy presents bio-stimulating properties related to the wavelength used [38]. 

### 2.7. Carboxytherapy

Carboxytherapy is the transdermal administration of carbon dioxide for therapeutic needs. Depending on the area of the body, the amount of CO_2_ infused is, respectively, 50–100 mL for the chin and arm, 200–300 mL for the thigh, and 300–600 mL for the abdomen. However, the role of carboxytherapy in lipolysis is still controversial. The local application of CO_2_ can enhance the peripheral circulation, improve tissue perfusion, increase the oxygen partial pressure through reflex vasodilation, and stimulate neoangiogenesis. An important aspect that proves the effectiveness of therapy in obesity treatment is based on the clear shrinkage of adipocytes, and a density decrease in the area exposed to carboxytherapy [39]. Brandi et al., in their study, observed histological changes after the local application of CO_2_, and noted damage to fat cells with the subsequent release of triglycerides in the intracellular spaces, which may be crucial in the development of this therapy [40]. 

### 2.8. High-Intensity Focused Electromagnetic Field

High-intensity focused electromagnetic field (HIFEM) is a non-invasive method used to reduce body fat, along with aiding in muscle repair. The device generates an electromagnetic field that activates the neuromuscular tissue through an induced electrical current, leading to muscle contractions. The significant energy demand forces the muscles to use the energy stored in adipocytes, i.e., free fatty acids. This consequently reduces the size and, when under an extreme muscle strain, the number of fat cells [41,42]. Studies show an average reduction in the abdominal circumference of 4.37 cm [43]. It is also possible to combine HIFEM with RF. Synchronized RF generates electromagnetic energy that can be regulated to concentrate mostly in the adipose tissue, leading to the selective heating of adipocytes in the temperature range of 42 °C to 45 °C, inducing apoptosis. Patients treated with HIFEM+ show an average of 30% less body fat, and 25% more muscle [41,44]. 

### 2.9. Pharmacological Methods

As well as the non-invasive methods that induce body fat reduction, there are a large number of pharmaceuticals that can also accelerate weight loss. Selected examples of drugs that affect human metabolism are presented below. 

Liraglutide and semaglutide (glucagon-like peptide-1) are used in type 2 diabetes, as well as in the treatment of obesity. Depending on the concentration of glucose in the blood, they regulate the pancreatic secretion of insulin and glucagon. These drugs are GLP-1 receptor agonists, which increase satiety and reduce hunger, resulting in a food intake reduction, and weight loss [4]. GLP-1 is an incretin hormone secreted in the intestines in response to food intake. It reduces the postprandial glucose levels by stimulating insulin secretion, and reducing glucagon secretion. It also suppresses the appetite, slows down gastric emptying, and reduces appetite and food intake. It has a positive effect on weight loss, and reduces the incidence of cardiovascular events. Biochemically modified analogs increase its half-life and potency [45]. GLP-1 receptors are found in many organs, including the brain, mainly in the hypothalamus, brainstem, and septal nucleus. Studies have shown that both drugs affect receptors in the brain, and will reduce the caloric intake. Semaglutide has a superior brain uptake compared to liraglutide, and the holistic effect is that semaglutide is twice as effective as liraglutide [46]. A study in rodents has shown that a central or peripheral administration of GLP-1R agonists reduces short-term food and water intake, and results in weight loss [47]. In a randomized trial, participants were randomly divided into four groups, in which the effects of liraglutide, in combination with exercise and a low-calorie diet, were studied for one year. At the endpoint, participants’ body weight was lowered by an average of 12% [48]. 

Through analyzing the many available methods to reduce the patient’s weight (Table 1), e.g., through the elimination of fat cells, we surmise that an ideal solution that will help patients achieve the intended effect in an accessible and minimally invasive way is still being sought.

## 3. Methods of the Future

### 3.1. Beige Adipocytes 

An innovative and physiology-based strategy is to stimulate the functional biogenesis of beige adipocytes, which occurs through cold and β3-adrenergic stimulation [49]. ADRB3 acts as an inductor of thermogenesis, and causes the dissolution of triglycerides, which results in an increase in insulin in the blood, and an increase in energy expenditure. ADRB3 is affected by leptin, glucocorticoids, insulin, and short-chain fatty acids, and its antagonists are adrenaline, noradrenaline, isoprenaline, and the currently unapproved drug BRL-37344 [10,50]. Likewise, bone morphogenetic proteins 4, 7, and 8b, vestibular and cerebral natriuretic peptides, fibroblast growth factor 21, vascular endothelial growth factor-α, and prostaglandins have been shown to promote cells becoming beige in vivo [51]. When the external excitatory stimuli are withdrawn, the mitochondria-enriched beige adipocytes transform into dormant adipocytes that resemble white adipocytes. In an experiment in mice, with a second cold stimulation, approximately half of the beige adipocytes in the WAT of the groin were from dormant cells, and half were from de novo precursor differentiation [10]. These observations could be used to create an effective mechanism based on beige adipocytes that would serve to reduce the volume of fat cells, and thus slim the waistline. However, the development of ADRB3 ligands for the treatment of obesity and metabolic diseases may have undesirable effects over time. Many clinical studies among humans have been conducted, which showed that a systematic exposure to low temperatures for several hours resulted in an increase in the activity of the adipose tissue, and weight loss. Before the possible clinical use of beige adipocyte induction, it should be assessed whether the metabolic effects are affected by the increase in UCP1, if there are alternative metabolic pathways, and if there is a more specific detector for the presence of beige adipocytes in the body (currently F-FDG-PET, F-fluorodeoxyglucose positron emission tomography) [51,52]. 

### 3.2. The Gene Encoding Myostatin and Interleukin-6 

A study in pigs looking at the effects of myostatin (MSTN) and interleukin 6 (IL-6) on metabolism also produced interesting findings. MSTN is mainly expressed in the muscles, but it also regulates fat accumulation. A lack of the MSTN gene results in excessive muscle growth (especially observed among children), and a reduced fat deposition. The experiment showed that the conversion of WAT to beige adipose tissue was not directly related to MSTN deficiency, but indirectly affected muscle IL-6 secretion. The infusion of IL-6 in healthy individuals stimulates lipolysis and fatty acid oxidation. Studies have shown that the skeletal muscle increases IL-6 production after physical exercise, and IL-6 secreted by the skeletal muscle leads to the burning of visceral fat [53,54]. 

### 3.3. Dopamine 

One of the more recent findings is the fact that dopamine D1 receptor (DRD1) signaling stimulates lipolysis and WAT browning [55]. This is due to the fact that one of the pathways affecting adipose tissue is the dopaminergic one, although the role of the dopamine receptors here is still unclear—what has been proven is that analogs can regulate adipocyte metabolism [56]. A team of researchers from Sichuan and Henan, China, proved the browning of adipocytes in mice in vitro and in vivo after the administration of fenoldopam (DRD1 agonist), because of a complex mechanism involving cyclic adenosine monophosphate and UCP-1 [55]. 

### 3.4. Curcumin and Anthocyanins 

Researchers have also begun to explore more natural pigments, such as curcumin and anthocyanins, which, by stimulating the secretion of glucagon-like peptide-1 (GLP-1), and inducing the formation of beige adipocytes, help in treating obesity and diabetes [57,58]. After using blackcurrant extract, which naturally contains delphinidin 3-rutinoside (D3R), in mice and rats, a reduction in blood glucose levels, and an improvement in glucose tolerance were observed, and juices with this composition reduce the postprandial blood glucose concentration in humans. The mechanism remains unclear, but it is likely due to the stimulation of GLP-1 by D3R. Importantly, D3R is not significantly broken down in the gastrointestinal tract for 45–60 min when taken in the form of the aforementioned extract. Curcumin, on the other hand, administered in the form of a specimen with high bioavailability (disperse), induces the formation of beige adipocytes. In addition, combining curcumin with artepillin C, which is a compound found in the world-famous Brazilian green propolis, significantly significantly induces the beige tissue formation, and allows the use of lower doses of both compounds [57]. 

### 3.5. GABA Receptors

The effects of gamma-aminobutyric acid (GABA) on obese mice fed a high-fat diet, and their gut microbiota composition, were investigated [59]. It has already been shown that the types and amount of bacteria in the intestines play an important role in metabolism, and have an impact on the formation and development of obesity [60]. This factor is easily modifiable by changing diet, using prebiotics, probiotics, antibacterial drugs, and fecal microbiota transplants. In one of the studies, GABA was used to investigate the mechanism in mice by which it affects the intestinal microflora, and improves the condition of the inguinal white adipose tissue (iWAT). The results showed that the use of GABA primarily changes the composition of the intestinal microflora (with an increased abundance of *Bacteroidetes, Akkermansia* and *Romboutsia*, and decreased *Firmicutes* and *Erysipelatoclostridium* in obese mice), reduces the body weight and adipose tissue inflammation, and promotes the expression of the thermogenic genes in iWAT. GABA treatment also increased the concentration of ketone bodies and adipocyte beige, and decreased the lipid oxidation. It can therefore be concluded that GABA has a significant effect on reducing body fat, and is a non-invasive method of reducing the body circumference [59]. 

### 3.6. Ephedra sinica Stapf 

The herbal extract *Ephedra sinica* (*E. sinica*), also known as Ma-Huang, is used in traditional Asian medicine to treat asthma, cough, and fever. The mechanism of the effect on adipose tissue is due to the active substance contained in it: ephedrine, i.e., an ADRB3 agonist [61]. Studies on mice using *E. sinica* have proven that it inhibits the accumulation of lipids and the expression of adipogenic genes; i.e., it also prevents the formation of new WAT cells. In addition, it causes an increase in the expression of UCP1 mRNA and protein, which results in adipocyte thermogenesis, and visible browning features. Beige and brown adipocytes have multiple mitochondria and a high expression of UCP1. Treatment with *E. sinica* results in an increased number and activity of mitochondria in WAT, thereby increasing cellular respiration and energy expenditure. The eventual desired effect is weight loss. In addition, researchers came to similar conclusions using this preparation in humans [62]. 

### 3.7. Adenosine 

Adenosine receptors are expressed in the adipose tissue, and control processes such as lipolysis and inflammation [63]. Endogenously released adenosine inhibits adipocyte lipolysis, but its analogs do not affect this process. In one study, adenosine deaminase, which degrades adenosine, was added to rat cell culture, causing lipolysis at peak levels [64]. However, the results of imaging studies in humans indicate that adenosine, in addition to the aforementioned effects, and lowering the concentration of free fatty acids, glycerol, and triglycerides in plasma, is also important in the activation of BAT [65]. In other studies, adenosine has been shown to promote alternative macrophage activation, prevent adipose tissue inflammation, and play a role in maintaining glucose homeostasis [66]. Taking into account its broad-spectrum effect, it should be considered as an active substance bound to adenosine receptors that would be most effective in reducing body weight in humans, but also harmless regarding the proper functioning of the organism [63,64,65,66].

### 3.8. Rhubarb 

The properties of rhubarb that help in the slimming process come from the mechanisms that lower the concentration of triglycerides, inhibit inflammation, and moderate the intestinal microbiota. Most of the pharmacological actions are derived from hydroxyanthraquinones (HAQs). Aloe emodin and rhein are the best-studied HAQs that work as inhibitors of adipocyte differentiation, and thus protect against obesity. To confirm the thesis, studies on mice with 3T3-L1 cells were carried out. The lipid accumulation was reduced by 32% and 38% after treatment with 50 μM emodin and rhein. The intracellular triglyceride levels were also markedly reduced with emodin and rhein intervention in differentiated 3T3-L1 cells. Rhubarb compounds inhibit adipogenesis, and increase lipolysis in 3T3-L1 cells, by controlling the expression of transcription factors, fatty acid synthase (FAS), acetyl-CoA carboxylase (ACC), and lipases. A special diet in rats led to weight gain, dyslipidemia, obesity, kidney dysfunction, and fatty liver, while the intraperitoneal administration of emodin or rhein alleviated all of these obesity-related symptoms. Compared to emodin, rhein showed a better therapeutic effect; therefore, further research on its properties in the treatment of obesity is recommended [67]. 

### 3.9. Cooling Clothes 

There is a growing need for non-invasive strategies for the treatment of obesity, which is why solutions are being sought to achieve this goal. Hanna Luze and colleagues began researching a new idea, which was cooling clothing. Subjects wore a gentle cooling belt around their hips and jaw for four weeks (one hour a day). The cooling effect is activated by holding the belt under tap water. The technologically advanced material retains water inside the garment, while the outer layer stays dry. The standard body temperature causes the evaporation of the water molecules bound in the material, which results in cooling by approximately 18 °C to 20 °C. After four weeks, a reduction in body weight by 0.9% was observed among the examined women, the BMI decreased by 0.8%, and the abdominal circumference by 1.7%. This is a novel method that needs further clinical trials to refine its long-term efficacy among patients [52].

### 3.10. Acupuncture 

The mechanism of acupuncture is associated with the lipolysis of adipose tissue, and thermogenesis. It is an effective and safe way to deal with obesity. In a rodent study, it was shown that electroacupuncture stimulation (ES) induces anti-inflammatory effects through autonomic nervous action. ES has also been confirmed to affect obesity, by inhibiting the noradrenaline transporter protein SlC6a2 in macrophages, and activating thermogenesis through the activity of the sympathetic nervous system. By inhibiting inflammasome activation, NLRP3 ES LI11 increased lipolysis. The ES LI11 was also shown to reduce body weight and pro-inflammatory cytokines, such as IL-6 and TNF-α, but did not promote anti-inflammatory cytokines, such as IL-4 [68,69]. 

### 3.11. CBL-514

CBL-514 is a new drug used in injection lipolysis. In an animal study, it was shown to induce apoptosis in adipocytes, and lipolysis in vitro. It also reduced subcutaneous fat in the injection areas. A phase IIa study was also conducted on 43 volunteers, in which they were randomly assigned to three designated dose levels of CBL-514: 1.2; 1.6, and 2.0 mg/cm^2^. After a 6-week treatment period, a follow-up was undertaken at 4 and 8 weeks post-treatment. All doses of CBL-514 were shown to reduce the absolute and percentage body fat, compared to the baseline. At the 2.0 mg/cm^2^ dose, the least-square-mean change from the baseline in the body fat thickness was −7.39 mm pr at the first follow-up, and −6.54 mm at the second follow-up (*p* < 0.00001). CBL-514, reducing the volume of adipose tissue in the abdominal area by 24.96%, has become a promising solution in non-surgical fat reduction, as well as a safe, well-tolerated therapy. However, further studies, with a larger group of participants, and the introduction of a control group, are required to obtain more reliable treatment results [70].

## 4. Conclusions

Due to the scale of the problem, it is necessary to constantly improve existing therapies to reduce body fat, as well as to search for new methods that will reduce the number of people affected by this disease. Potential can be seen in substances used in the treatment of obesity, which include: rhubarb, curcumin and anthocyanins, as well as adenosine, and a new drug used in injection lipolysis: CBL-514. The ongoing research based on physiological mechanisms of stimulating the functional biogenesis of beige adipocytes is also an innovative and promising approach that offers future solutions.

## Figures and Tables

**Table 1 medicina-59-01378-t001:** Advantages and disadvantages of current methods of fat removal.

Current Methods	Advantages	Disadvantages
Cryolipolysis	The effects are visible after one session, safety profile: side effects insignificant.	Skin pulling and pinching during treatment, the effects take some time (3–4 months) to show.
Laser lipolysis	Causes the apoptosis of adipocytes via micropores in their membranes, cooling system to protect against the side effects, additionally removes skin imperfections and signs of skin aging, and lowers cholesterol and leptin levels, a safe method.	Skin burning, edema, pigmentation, collagen denaturation, loss of hair follicles, disappearance of the adipose layer and damage to the underlying muscle (side effects occur especially in the absence of cooling).
Injection lipolysis	Natural active substances such as phosphatidylcholine or sodium deoxycholate cause cell death, migration and stimulation of fibroblasts, cheap and easily available.	Pain, edema, bruising, in rare cases, skin ulceration, eventually the accumulation of more collagen, and an uneven distribution of adipose tissue, the effects take some time (3–4 months) to show.
Ultrasonic lipolysis	Completely non-invasive and painless, reduces the abdominal cellulite deposition, no need for a recovery period.	Not always effective, possiblly an insignificant amount of regression.
Radiofrequency lipolysis	Firms the skin.	Gently reduces body fat (more useful as an additional therapy with other lipolysis).
Low-Level Laser Therapy	Fat and cellulite reduction, without any significant side effects and without pain.	Gently reduces body fat (more useful as an additional therapy with other lipoplasty).
Carboxytherapy	Additionally used for dark circles and puffiness under the eyes, hair loss, stretch marks, anti-aging prophylaxis, safety profile.	Ecchymosis and pain, the minimum series consists of six sessions.
High-intensity focused electromagnetic field	Reduces body fat while increasing muscle mass, painless, no serious side effects, no inflammatory reactions.	Effects after three months, muscle pain, periodic muscle cramps, periodic joint or tendon pain.
Pharmacological methods—liraglutide and semaglutide	Slows down gastric emptying, and reduces appetite and food intake, reduces the incidence of cardiovascular events, effective in the treatment of obesity and diabetes.	Expensive drugs for monthly treatments, nausea and diarrhea (especially at the beginning), hypoglycemia (during combination therapy with insulin or a sulfonylurea).

## Data Availability

Not applicable.

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
