# Peer review of "Modern and Non-Invasive Methods of Fat Removal"

_medicina, 2023, doi:10.3390/medicina59081378_

Round 1
Reviewer 1 Report
This article reported modern and non-invasive methods of fat removal, which is an interesting topic. However, there are some minor concerns in this article that need to be addressed.
1. In the first citation, the abbreviation for the word should be preceded by the full name, and then write the abbreviation (e.g. BMI, GLP1, BRL, etc.). Please correct it.
2. Please correct the range of BMI for overweight.
3. Please include appropriate references at the end of sentences.
“Obesity is defined as abnormal or excessive fat accumulation that may impair health. Adipose tissue, otherwise known as body fat, is a type of connective tissue in which fat storage cells predominate. It consists mainly of cells called adipocytes, which are formed from fibroblast-like, progenitor, undifferentiated cells - preadipocytes.”
“Brown Adipose Tissue (BAT) plays a main role in energy homeostasis and thermogenesis among mammals. It prevents obesity caused by diet and hypothermia throughout the Uncoupling Protein 1 (UCP1). This mechanism is very important in young children due to the undeveloped capacity of thermoregulation.”
“Nowadays, more than 40% of the calories in a typical diet of a modern human in developed countries are supplied with fat, and in a similar amount with carbohydrates, which, supplied in excess, are converted into triglycerides”
“Laser lipolysis has been a commonly used method to reduce body fat since 2006. The generated laser beam with possible wavelengths of 924, 968 and 980 nm for the diode laser and 1064, 1319, 1320 and 1440 nm for the Nd:YAG laser (neodymium-yag laser) penetrates the tissue, is scattered, reflected and absorbed.”
“As a result of subcutaneous injection of PPC into adipose tissue, adipocytes burst and phosphatidylcholine increases the secretion of triglyceride-rich lipoproteins. DC is a component of human bile acid which helps in the emulsification and digestion of fats in the intestines. However, when administered externally, it damages the cell membrane of adipocytes, causing their death”
“One of the more recent findings is the fact that dopamine D1 receptor (DRD1) signal- 260 ing stimulates lipolysis and WAT browning.”
“Researchers have also begun to explore more natural pigments such as curcumin and anthocyanins, which, by stimulating the secretion of glucagon-like peptide-1 (GLP-1) and inducing the formation of beige adipocytes, helps in treating obesity and diabetes”
“The effects of gamma-aminobutyric acid (GABA) on obese mice fed a high-fat diet and their gut microbiota composition were investigated. It has already been shown that the types and amount of bacteria in the intestines play an important role in metabolism and have an impact on the formation and development of obesity”
“The herbal extract E. sinica, also known as Ma-Huang, is used in traditional Asian medicine to treat asthma, cough, and fever. The mechanism of the effect on adipose tissue is due to the active substance contained in it - ephedrine, i.e. an ADRB3 agonist.”
4. Instead of assigning multiple references for each paragraph, each sentence should have one reference.
5. Please explain more about pharmacological methods such as agonists and antagonists of other hormones.
6. Please change CO2 to CO2.
7. Please provide the full name of E. sinica in the manuscript.
8. For better understanding, please compare advantages and disadvantages of these methods in table.
9. I would suggest including a figure for the “Methods of the future” section.
10. The name of genes and bacteria are not in italics. Please correct them.
Minor editing of English language required
Author Response
We thank the reviewer for his thoughtful and thorough review and believe his input has been invaluable to make our manuscript more balanced. Below are answers to these suggestions.
This article reported modern and non-invasive methods of fat removal, which is an interesting topic. However, there are some minor concerns in this article that need to be addressed.
- In the first citation, the abbreviation for the word should be preceded by the full name, and then write the abbreviation (e.g. BMI, GLP1, BRL, etc.). Please correct it.
Thank you for your attention. Corrected.
- Please correct the range of BMI for overweight.
Thank you for that. done.
- Please include appropriate references at the end of sentences.
“Obesity is defined as abnormal or excessive fat accumulation that may impair health. Adipose tissue, otherwise known as body fat, is a type of connective tissue in which fat storage cells predominate. It consists mainly of cells called adipocytes, which are formed from fibroblast-like, progenitor, undifferentiated cells - preadipocytes.”
“Brown Adipose Tissue (BAT) plays a main role in energy homeostasis and thermogenesis among mammals. It prevents obesity caused by diet and hypothermia throughout the Uncoupling Protein 1 (UCP1). This mechanism is very important in young children due to the undeveloped capacity of thermoregulation.”
“Nowadays, more than 40% of the calories in a typical diet of a modern human in developed countries are supplied with fat, and in a similar amount with carbohydrates, which, supplied in excess, are converted into triglycerides”
“Laser lipolysis has been a commonly used method to reduce body fat since 2006. The generated laser beam with possible wavelengths of 924, 968 and 980 nm for the diode laser and 1064, 1319, 1320 and 1440 nm for the Nd:YAG laser (neodymium-yag laser) penetrates the tissue, is scattered, reflected and absorbed.”
“As a result of subcutaneous injection of PPC into adipose tissue, adipocytes burst and phosphatidylcholine increases the secretion of triglyceride-rich lipoproteins. DC is a component of human bile acid which helps in the emulsification and digestion of fats in the intestines. However, when administered externally, it damages the cell membrane of adipocytes, causing their death”
“One of the more recent findings is the fact that dopamine D1 receptor (DRD1) signal- 260 ing stimulates lipolysis and WAT browning.”
“Researchers have also begun to explore more natural pigments such as curcumin and anthocyanins, which, by stimulating the secretion of glucagon-like peptide-1 (GLP-1) and inducing the formation of beige adipocytes, helps in treating obesity and diabetes”
“The effects of gamma-aminobutyric acid (GABA) on obese mice fed a high-fat diet and their gut microbiota composition were investigated. It has already been shown that the types and amount of bacteria in the intestines play an important role in metabolism and have an impact on the formation and development of obesity”
“The herbal extract E. sinica, also known as Ma-Huang, is used in traditional Asian medicine to treat asthma, cough, and fever. The mechanism of the effect on adipose tissue is due to the active substance contained in it - ephedrine, i.e. an ADRB3 agonist.”
Thank you for this attention. Corrected. We added citations that were missing or changed the order of citations.
- Instead of assigning multiple references for each paragraph, each sentence should have one reference.
Thank you for this attention. I corrected the most visible place
- Please explain more about pharmacological methods such as agonists and antagonists of other hormones.
Thank you for that. We added.
- Please change CO2 to CO2.
Thank you for your attention. Done.
- Please provide the full name of E. sinica in the manuscript.
Thank you for that. Done.
- For better understanding, please compare advantages and disadvantages of these methods in table.
Thank you for that advice. We made one.
- I would suggest including a figure for the “Methods of the future” section.
Thank you for this, but in this work we will not insert a picture, because we do not have the appropriate programs to create and we prefer not to insert anything so as not to diminish the work.
- The name of genes and bacteria are not in italics. Please correct them.
Thank you for your attention. Done.

Reviewer 2 Report
Interesting review paper on modern and non-invasive fat removal methods.
Narrative review based on 65 bibliographical references, of which 27 from the last 3 years.
Review of current methods and future perspectives of treatments.
Author Response
Thank you very much for your kind words and appreciation of our work. We are immensely grateful
Reviewer 3 Report
The authors through their perspective have described the modern non-invasive method of fat removal with insights on several other harmless solutions that can be incorporated or combined with the present leading techniques to reduce obesity worldwide. The authors along with the five leading non-invasive techniques of fat removal have described other current methods like carboxytherapy, high-intensity focused electromagnetic field therapy, pharmacological methods and low-level laser therapy which combined with liposuction can help patients achieve a reduction in weight in a minimally invasive way without any side effects. The authors have also mentioned other methods that can be used as future therapies for fat reduction by the browning of white adipose tissue by targeting myostatin, interleukin-6 or use of dopamine D1 receptor agonist, β-3 adrenergic agonist, adenosine analogs, gamma-aminobutyric acid, curcumin, etc. However herbal extract like that from Ephedra sinica and rhubarb need more testing as they have toxicity and is not approved in many countries and thus cannot be used as a safe method for the treatment of obesity which the authors are suggested to mention in their manuscript. Overall, the perspective is simple to understand and describes the non-invasive methods of fat removal efficiently. However, I would like to suggest the authors address a few minor revisions before the perspective is considered for publication.
The authors are recommended to correct the minor grammatical errors particularly in the abstract and typos by revising them through the MDPI language editing service or any other proofreading services.
Please correct the typo in line 185.
The authors are suggested to check lines 196-204 of the manuscript as they have written the sentence twice.
The authors may add schematic representations of the methods they have mentioned to make the perspective more appealing to the readers.
As mentioned in the reviewers' comments the authors are suggested to use proofreading services for their manuscript.
Author Response
We thank the reviewer for his thoughtful and thorough review and believe his input has been invaluable to make our manuscript more balanced. Below are answers to these suggestions.
The authors through their perspective have described the modern non-invasive method of fat removal with insights on several other harmless solutions that can be incorporated or combined with the present leading techniques to reduce obesity worldwide. The authors along with the five leading non-invasive techniques of fat removal have described other current methods like carboxytherapy, high-intensity focused electromagnetic field therapy, pharmacological methods and low-level laser therapy which combined with liposuction can help patients achieve a reduction in weight in a minimally invasive way without any side effects. The authors have also mentioned other methods that can be used as future therapies for fat reduction by the browning of white adipose tissue by targeting myostatin, interleukin-6 or use of dopamine D1 receptor agonist, β-3 adrenergic agonist, adenosine analogs, gamma-aminobutyric acid, curcumin, etc. However herbal extract like that from Ephedra sinica and rhubarb need more testing as they have toxicity and is not approved in many countries and thus cannot be used as a safe method for the treatment of obesity which the authors are suggested to mention in their manuscript. Overall, the perspective is simple to understand and describes the non-invasive methods of fat removal efficiently. However, I would like to suggest the authors address a few minor revisions before the perspective is considered for publication.
The authors are recommended to correct the minor grammatical errors particularly in the abstract and typos by revising them through the MDPI language editing service or any other proofreading services.
Please correct the typo in line 185.
Thank you for your attention. done.
The authors are suggested to check lines 196-204 of the manuscript as they have written the sentence twice.
Thank you for this attention. Corrected.
The authors may add schematic representations of the methods they have mentioned to make the perspective more appealing to the readers.
Thank you for this, but in this work we will not insert a picture, because we do not have the appropriate programs to create and we prefer not to insert anything so as not to diminish the work. Instead of picture, we added a table.
Thank you again for reviewing our work and we hope that this new version is satisfactory.
Yours faithfully
Authors
